

# Growth and actual leaf temperature modulate CO₂-responsiveness of monoterpene emissions from Holm oak in opposite ways

Michael Staudt[1], Juliane Daussy[1], Joseph Ingabire[1], Nafissa Dehimeche[1]

[1] CEFE, CNRS, EPHE, IRD, Univ Montpellier, Montpellier, France

*Correspondence to*: Michael Staudt (michael.staudt@cefe.cnrs.fr)

**Abstract.** Climate change can profoundly alter VOC emissions from vegetation and thus influence climate evolution. Yet, the short and long-term effects of elevated $CO_2$ concentrations on emissions in interaction with temperature are not enough understood, especially for VOCs other than isoprene. We measured $CO_2$-response curves of leaf monoterpene emissions and photosynthetic parameters at two assay temperatures (30 and 35°C) on four populations of holm oak saplings grown under

normal and double $CO_2$ concentrations combined with two temperature growth regimes differing by 5 °C. A stepwise reduction in $CO_2$ resulted in a decrease in emissions, occasionally preceded by an increase, with the overall decrease in emissions being greater at 35 °C than at 30 °C assay temperature. During ramping to high $CO_2$, emissions remained mostly unchanged at 35 °C, whereas at 30 °C they often dropped, especially at the highest $CO_2$ levels (≥ 1200 ppm). In addition to the actual leaf temperature, the high $CO_2$-responsiveness of emissions was modulated by the plant's growth temperature with warm-grown

plants being more sensitive than cool-crown plants. In contrast, growth $CO_2$ had no significant effect on emission $CO_2$ sensitivity, although it promoted plant growth and the leaf's emission factor. Correlation analyses suggest that the emission response to low $CO_2$ depended on the leaf's initial carbon balance and its actual energy status, whereas the response to high $CO_2$ depended only on the leaf's actual energy status, which was affected by the occurrence of photooxidative stress and feedback limitation of photosynthesis. Overall, our results confirm an isoprene-analogous behavior of monoterpene emissions

from holm oak. Emissions exhibit a nonlinear response curve similar to that currently used for isoprene emission in the MEGAN model, with no difference between major individual monoterpene species and plant chemotype. Simulations estimating annual VOC releases from holm oak leaves at double atmospheric $CO_2$ indicate that the observed high-$CO_2$ inhibition is unlikely to offset the increase in emissions due to expected warming.

## 1 Introduction

Terrestrial vegetation has been identified as the main source of biogenic volatile organic compounds (BVOCs). Besides their multiple biological functions, BVOCs influence several climate forcing components in the atmosphere, notably the concentrations of the greenhouse gases methane and ozone, and in pristine environments the formation of secondary organic aerosols (Fuentes et al., 2001; Arneth et al., 2010). Aerosols can have a cooling effect on the earth climate by increasing the diffusive fraction of radiation and by changing cloud properties in the atmosphere (Zhu et al., 2019; Yli-Juuti et al., 2021).



Furthermore, increased diffusive light can favor photosynthesis possibly enhancing carbon sequestration in forest ecosystems
(Ezhova et al., 2018; Rap et al., 2018). Consequently, large-scale alteration of BVOC emissions due to global change could
feedback on future climate evolution (Scott et al., 2018; Sporre et al., 2019). Yet, the interacting effects of climate change
factors on emissions are currently not enough understood. Among these, increasing atmospheric concentrations of carbon
dioxide [$CO_2$] and temperature are major factors with no or relative weak regional differences compared to other factors such

as precipitation.

Globally, volatile isoprenoids constitute the largest fraction of BVOCs (Guenther et al., 2012). Isoprene alone accounts about
the half, which is produced at high rates in the photosynthesizing tissues of about 30% of vascular plant species with a higher
presence in woody species than in herbs (Monson et al., 2012; Fineschi et al., 2013, Sharkey et al., 2013; Dani et al., 2014a).
Accordingly, most studies have focused on isoprene emissions (for recent reviews see Sharkey and Monson, 2014; Lantz et

al., 2019a; Monson et al. 2021). Isoprene is synthesized in chloroplasts from dimethylallyl diphosphate (DMADP) by isoprene
synthase, an enzyme with relatively low affinity for its substrate (Lehning et al., 1999; Sharkey et al., 2013). Isoprene synthesis
and emissions are strongly regulated by temperature and light, which is mainly related to changes in the pool size of DMDAP.
DMADP is built in the chloroplastic methyl-D-erythritol-4-phosphate (MEP) pathway from the C3-substrates glyceraldehyde-
3-phosphate (triose-phosphate) and pyruvate, plus reduction power (NADPH or equivalents) and phosphorylation power (ATP

or equivalents) as energetic cofactors. Glyceraldehyde-3-phosphate and energetic cofactors come directly from ongoing
photosynthesis, while a large fraction of pyruvate is formed from phosphoenolpyruvate, which is imported from the cytosol in
exchange of inorganic phosphate (Pi) and thus can stem from "older" carbon sources (Lantz et al. 2019a and references therein).
Regarding $CO_2$-effects, many studies observed that isoprene emissions decrease rapidly at high [$CO_2$] (e.g., Monson and Fall,
1989; Loreto and Sharkey, 1990; Monson et al., 1991; Rasulov et al., 2009; Wilkinson et al., 2009; Possell and Hewitt, 2011;

Morfopoulos et al., 2014), potentially counteracting the expected increase in emissions from rising temperatures and $CO_2$
fertilization on plant growth in a future warmer, high-$CO_2$ world (Pacifico et al., 2012; Bauwens et al. 2018). This $CO_2$-
responsiveness however can be variable from one species to another (e.g. Sharkey et al., 1991; Lantz et al., 2019b; Niinemets
et al., 2021), and be modulated by the atmospheric $CO_2$-regimes, in which the plants were grown or acclimated (e.g. Wilkinson
et al., 2009; Possell and Hewitt, 2011; Sun et al., 2013). Further, it depends on the actual leaf temperature; high temperatures

generally suppress the high-$CO_2$ sensitivity of isoprene emissions (e.g. Affek and Yakir, 2002; Rasulov et al., 2010; Sun et al.,
2013; Potosnak et al., 2014, Dani et al., 2014b; Monson et al., 2016).

Monoterpenes (MTs) are less emitted globally (<15%; Guenther et al., 2012). However, in some vegetation types such as
boreal, temperate mountainous and Mediterranean forests, MTs can largely dominate the total BVOC release (e.g. Rantala et
al., 2015; Seco et al., 2017; Tani and Mochizuki, 2021 and references therein), and due to their bigger size and high reactivity,

these emissions might be particular relevant for local and regional SOA formation (Jokinen et al., 2015; Zhang et al., 2018;
McFiggans et al., 2019). MTs are essentially produced in the same pathway as isoprene (but see Pazouki and Niinemets, 2016
for exceptions). However, the responses of MT emissions to elevated [$CO_2$] are less understood and show more contrasting
results ranging from no effect, increases and decreases (Arneth et al., 2008; Peñuelas and Staudt, 2010; Feng et al., 2019;





Daussy and Staudt, 2020). There are several reasons why MT emissions may behave differently than isoprene. First, in many

plants MTs are synthesized outside photosynthetic source tissues in glandular organs (trichomes, resin ducts), where there can be accumulated in high concentrations (e.g. Huang et al., 2018; Dehimeche et al., 2021). As a result, emission rates vary independent from their biosynthesis rates, which furthermore, is less coupled to ongoing photosynthesis due to additional regulatory processes associated with the partitioning and transport of photosynthates. Second, the biosynthesis of MTs involves at least two other enzymes (i.e. one MT synthase and one geranyl diphosphate synthase) whose in-planta catalytic rates may

be less substrate regulated than isoprene due to their high substrate affinities (Harrison et al., 2013; Rasulov et al., 2014 and references therein). In addition, MT emitters typically produce several MTs formed by several MT synthases thus introducing further complexity in their responses to $[CO_2]$.

One of the best studied MT emitter is the Mediterranean evergreen oak *Quercus ilex* L. (QI, holm oak). Its strong MT emissions show much analogy to isoprene in terms of quantity and responses to environmental factors (Loreto et al., 1996a; Staudt and

Bertin, 1998). Regarding $CO_2$ effects on emissions, two studies reported that emissions become significantly inhibited at high very $[CO_2]$ but not at moderately increased $[CO_2]$ (Loreto et al., 1996b; Staudt et al., 2001). Yet, Loreto et al., (2001) compared emissions from QI trees growing in open top chambers with normal and double $[CO_2]$ and concluded that 700 ppm $[CO_2]$ significantly inhibits the emissions some of the major MTs while enhancing others in consistence with differences in the tree's activity of MT synthases. However, because in that study emissions at different $CO_2$ levels were determined on different trees,

the seeming compound specificity could be confounded with the chemotype of the tree, a possible misinterpretation mentioned by the authors. Later, Rapparini et al. (2004) investigated emissions from QI trees growing near natural $CO_2$ springs. Switching from 350 ppm to 1000 ppm reduced all emissions in the control site but not in the elevated $CO_2$ site. They also found unexplained seasonal differences in $CO_2$ responsiveness possibly associated with water stress. Long-term, seasonal $CO_2$ effects were also reported by Staudt et al. (2001) who observed that the emission factor (EF, i.e. the foliar emission rate under standard

temperature, light and $[CO_2]$) of elevated $CO_2$-grown plants was significantly increased during the winter season but not before during the warm season. Thus, while previous studies provide evidence that MT emissions from QI can be inhibited by elevated $[CO_2]$, the exact $CO_2$-response, its compound specificity, and its dependence on actual temperature and growth conditions are unclear or not known. To gain additional insight, we conducted a study in which we measured $CO_2$ response curves of foliar MT emissions, $CO_2/H_2O$ gas exchanges, and chlorophyll fluorescence at two assay temperatures on four QI populations that

were grown under two $[CO_2]$ in combination with two temperature regimes. In particular, we addressed the following questions: Do MT emissions respond to low and/or high $[CO_2]$ and how are these responses related to the leaf's emission factor and primary metabolism? Does the $CO_2$-responsiveness differ between individual MT compounds? Is the $CO_2$-responsiveness affected by the actual leaf temperature? Do seedlings grown under warmer and/or higher $CO_2$ regimes differ in their leaf emission factors and/or responsiveness to $CO_2$? Are the observed effects relevant for estimating MT emissions in a future

warmer and $CO_2$-enriched world?



## 2 Material and methods

### 2.1 Plants and growth conditions

QI acorns were sampled in fall from various adult trees growing around Montpellier. They were stored in boxes with humidified paper tissues in a cold chamber at 5°C, where they started to germinate during winter and early spring. The germinating acorns were subsequently potted in PVC-pipes (diameter 16 cm, height 100 cm) containing a mix of sand, clay and peat, and were immediately transferred in four identical, controlled-environment greenhouse compartments. Temperature regimes and atmospheric [$CO_2$] of the compartments were set to 400 ppm $CO_2$ and 15/25 °C night/day temperature (400/20), 800 ppm $CO_2$ and 15/25 °C (800/20), 400 ppm $CO_2$ and 20/30 °C, (400/25) and 800 ppm $CO_2$ and 20/30 °C (800/25). Currently in the Montpellier region, a temperature range of 15/25 °C is common in late spring when new leaf growth is at its peak, whereas the 20/30 °C is typical for the summer (see table S1 in supplement 1). However, in a future atmosphere with 800 ppm [$CO_2$], such high temperatures are expected to occur more frequently outside the summer season especially in the Mediterranean, where climate warming proceeds faster than global average (Seneviratne et al., 2016). Plants were grown in these compartments under the same temperature and $CO_2$ regimes for 4-5 months until measurements started. The greenhouse facility of our institute consists of eight serial compartments in about east-west direction with the southern facades exposed to a large open field (grassland). To avoid any edge effect and uneven light exposure, the four inner compartments were used for the experiment. In addition, plants were regularly moved within the greenhouse compartments as well as between greenhouse compartments after having interchanged the growth temperature and $CO_2$ regimes.

### 2.2 $CO_2$-response curve measurements

Leaf MT emissions (E), $CO_2$/$H_2O$ gas exchange and chlorophyll fluorescence were measured using two LI-6400 Portable Photosynthesis Systems (LI-COR Biosciences, Lincoln, NE, USA). The large majority of measurements were made with the small 2 cm² Li6400-40 leaf chamber equipped with a blue/red LED light source and an integrated fluorometer. At the beginning of the experiment, few additional measurements were conducted with a 6 cm² broadleaf chamber equipped with a LED source but without fluorometer. For measuring the $CO_2$ response of VOC emission, a mature, healthy leaf of a sapling was gently clamped into the chamber and subsequently exposed to seven ambient $CO_2$ levels in the following order: 400, 200, 100, 800, 1200, 1600, 2000 ppm. The flow rate of air was set at 300 µmol s$^{-1}$ (ca. 450 ml min$^{-1}$), photosynthetic photon flux density (PPFD) at 1000 µmol m$^{-2}$ s$^{-1}$ (10% blue, 90% red LEDs) and block temperature at either 30 or 35 °C. Leaves were equilibrated at every $CO_2$ level for at least 30 min before starting data recording and VOC sampling. We applied a relative long waiting periods compared to analog studies on isoprene (e.g. Monson et al., 2016; Lantz et al., 2019b), because MT emissions need longer to come to a new steady state due to their lower volatility (Niinemets et al., 2002a; Staudt et al., 2003). BVOCs were sampled with a programmable air sampler (Gillian GilAir Plus, Sensidyne LP, USA) passing 2.4 L of chamber air at 150 ml min$^{-1}$ through adsorption cartridges packed with about 180 mg Tenax TA and 130 mg Carbotrap B. The chamber air was taken from the air hose connecting the chamber and match valve via a 3-way Teflon valve (Bola, Bohlender GmbH, Germany).



$CO_2/H_2O$ gas exchange and chlorophyll fluorescence data were recorded before and after VOC sampling surveying each time that gas exchange was stable. The mean values of both records were used for further data evaluation. All photosynthetic variables (net $CO_2$-assimilation (A), transpiration, conductance to water vapor (G), substomatal [$CO_2$] (Ci)) were calculated by the LiCOR software including corrections for diffusion leaks as recommended for small leaf chambers and if necessary for leaf area. In fact a few times, when the 6 cm$^2$ leaf chamber was used, the enclosed leaf did not completely cover the chamber surface. In that case, we marked the enclosed leaf part and measured separately its projected surface.

The integrated leaf chamber fluorimeter determined the actual quantum efficiency of photosystem II (PSII) electron transport in the light ($\Phi$PSII) by measuring first the steady-state fluorescence (Fs) of the light adapted leaf and then the maximum fluorescence (Fm') by applying a saturating light pulse of ca. 10000 μmol m$^{-2}$ s$^{-1}$ PPFD ($\Phi$PSII = (Fm'-Fs) Fm'$^{-1}$) (Murchie and Lawson, 2013). $\Phi$PSII is proportional to the flow of electrons in PSII (electron transport rate, ETR), which was calculated by multiplying $\Phi$PSII with PPFD assuming that 87% of the incident PPFD was absorbed by QI leaves, of which half is attributed to PSII (i.e. ETR = $\Phi$PSII * PPFD * 0.87 * 0.5). It should be noted that these correction factors may somewhat have varied among individual leaves and measurements, for example due to differences in the leaf's structure and light acclimation (Laisk and Loreto, 1996; Niinemets et al., 2006; McClain and Sharkey, 2019). However, leaf-to-leaf variation of calculated ETRs showed no significant relation to the variation of leaf structural variables (leaf mass per area (LMA), chlorophyll concentration ($R^2$ < 0.1, P > 0.15)). In addition to measurements under light, foliar dark respiration and maximum quantum efficiency of PSII photochemistry (Fv/Fm) were measured at the beginning and at the end of $CO_2$-ramping after having leaves adapted to dark and 400 ppm [$CO_2$] for 30 min. Fv/Fm is given as (Fm-Fo) Fm$^{-1}$, where Fo and Fm are, respectively, the steady-state and maximum fluorescence of the dark-adapted leaf. Fm and Fm' data were further used to calculate the non-photochemical quenching (NPQ) as (Fm'-Fm) Fm'$^{-1}$. NPQ reflects the fraction of absorbed light energy dissipated as heat from PSII. NPQ englobes protective mechanisms against excessive light, which otherwise leads to the over-reduction of PSII along with the formation of Reactive Oxygen Species (ROS: mainly singlet oxygen, superoxide, hydroxyl radicals and hydrogen peroxide; Asada, 2006) and ultimately to the persistent photoinhibition of the photosystem as indicated by a drop in Fv/Fm. NPQ processes are regulated by the acidification of the chloroplast thylakoid lumen leading to the activation of the integral membrane protein PsbS and the xanthophyll cycle, both triggering conformational changes in PSII antenna (Ruban, 2016).

Other derived variables considered in the data evaluation were the ETR/A, E/A and E/ETR ratios. ETR/A is the amount of electrons per net-assimilated $CO_2$. Variation in ETR/A reflects the excess of photochemical energy produced via PSII not used for $CO_2$ reduction in the Calvin-Benson-Bassham (CBB) cycle, hence the amount of NADPH and ATP available for other metabolic pathways inside chloroplasts such as photorespiration, starch synthesis, nitrite reduction, Mehler reaction (oxygen reduction), xanthophyll cycle and isoprenoid biosynthesis. Under physiological normal conditions, about half of the ETR is used for $CO_2$ reduction (Dani et al., 2014b), with about four to five moles of electrons required per assimilated mole of $CO_2$. The E/A and E/ETR ratios are respectively the percentage losses of assimilated carbon (C-loss) and PSII photosynthetic



electron transport (é-loss) by MT emissions assuming that one mole emitted MT consumes 10 moles of assimilated carbon and 56 moles of electrons (28 moles NADPH or equivalents and 2 moles electrons per NADPH; Niinemets et al. 2002b).

All response curves were run in the greenhouse compartments. The air for the LiCOR instrument was always taken from outside the greenhouse and filtered with charcoal to minimize $[CO_2]$ fluctuations and contamination with ambient VOC. The response curves at 30 and 35 °C were always measured on different mature leaves of a given sapling. Due to logistic constrains, the number of replicates per growth treatment varied between 5 and 8 at 30 °C and between 4 and 6 at 35 °C assay temperature (totally 26 $CO_2$-response curves at 30 °C and 20 $CO_2$-response curves at 35 °C on 26 saplings). We favored running more replicates at 30°C than at 35°C due to the stronger and more variable $CO_2$-responsiveness at this assay temperature. On the whole, one $CO_2$-response curve lasted about 6 hours. In order to check whether BVOC emissions from QI leaves changed during the day independently of external factors, we repeatedly measured emissions from QI leaves according the same protocol but without changing assay $[CO_2]$, temperature or PPFD.

Additional ancillary measurements were made after each experiment: Relative chlorophyll contents of the measurement leaves were assessed using a SPAD-502 instrument (Minolta, Ltd, Japan). SPAD data were converted to foliar Chlorophyll concentration ([Chloro]) based on the calibration realized in a previous study on QI seedlings (Staudt et al., 2017). Further, projected leaf area were determined by means of a scanner plus image software (Epson perfection V800; Image J5 software, National Institutes of Health, Bethesda, MD, USA) and dry weights on a microbalance after oven drying at 60 °C for 48 hours. Plant growth was assessed by measuring the number of leaves and ramifications and total plant height.

**2.3 BVOC analyses**

Adsorption Cartridges were analyzed using a gas chromatograph coupled with mass spectrometer (GC-MS) Shimadzu QP2010 Plus equipped with a Shimadzu TD-20 thermodesorber (Shimadzu, Kyoto, Japan). Prior to analysis, cartridges were purged for 1 min with dry $N_2$ at room temperature to remove excess water. BVOCs were thermally desorbed from cartridges at 250 °C in a 30 ml min$^{-1}$ He flow for 10 min on a cold trap filled with Tenax TA and maintained at - 10 °C. The focused VOCs were then thermally injected into the GC-column with a split ratio of 4 by flash heating the cold trap to 240 °C for 5 min. BVOCs were separated on a DB5 column (30 m x 0.25 mm, 0.25 μm film thickness) with helium as carrier gas (constant flow 1 ml min$^{-1}$) using the following oven temperature program: 2 min at 40 °C, 5 °C min$^{-1}$ to 200°C, 10 °C min$^{-1}$ to 270 °C held for 6 min. Eluting BVOCs were identified by comparison of mass spectra and arithmetic retention indices with commercial databases (NIST 2005; Wiley 2009; Adams 2005) as well as with commercial pure standards (Fluka, Sigma) dissolved in methanol to achieve realistic concentrations. Liquid standards stepwise dissolved in methanol were also used to calibrate the GC-MS system. The present study was focused on the five predominantly emitted MTs that were α-pinene, sabinene, β-pinene, myrcene and limonene. The emission rates were calculated by multiplying the chamber net BVOC concentration (i.e. chamber BVOC concentration with plant minus BVOC concentration of empty chamber) with the chamber flow rate divided by the enclosed leaf area, which in most cases was equal the chamber area (see above). Empty chamber was either measured before or after $CO_2$ -ramping. The emission rates per leaf dry weight were calculated using the LMA of the measured leaf.





## 2.4 Data treatments and statistical tests

During $CO_2$-ramping stomatal conductance and transpiration frequently increased at low $CO_2$ and diminished at high $CO_2$. As a result, leaf temperature slightly changed during $CO_2$-ramping, sometimes by much as 1°C owing to changes in evaporative cooling of the leaf by transpiration. To avoid that potential $CO_2$ effects on emissions were biased by leaf temperature changes, we normalized the emissions rate to the same standard temperature of 30°C (hereafter referred to as the emission factor (EF)) using the temperature algorithm for light dependent isoprenoid emission (Guenther et al., 1993) with coefficients adjusted for

QI emissions according to Staudt and Bertin (1998). Previous studies showed that the temperature response of MT emissions from QI closely resemble that assumed for isoprene emissions with limited variation due to plant acclimation and interaction with other environmental factors (Staudt and Bertin, 1998; Staudt et al., 2003).

To examine the relative changes of BVOC emissions and photosynthetic variables in response to $CO_2$-ramping, the data of each $CO_2$-response curve were normalized in two ways: i) By dividing the individual values of a measurement series by the

mean of the series ($X_{[CO2]} X_{mean}^{-1}$); ii) by dividing the individual values of a measurement series by the initial value, i.e. the measurement made at 400 ppm $CO_2$ ($X_{[CO2]} X_{400}^{-1}$). The first normalization is relatively insensitive to outliers and served to assess and illustrate the overall $CO_2$ responsiveness of a measured variable. However it is less suitable to differentiate the responsiveness to low $CO_2$ from that to high $CO_2$, because the response of one will influence the relative response of the other (see also Fig. S1 in Supplement 2 for illustration of the potential biases generated by data normalizations). Hence, the second

normalization was specifically used to analyze separately the responsiveness to low $CO_2$ (2 measurements at $[CO_2] < 400$ ppm) and to high $CO_2$ (4 measurements at $[CO_2] > 400$ ppm) of a measured variable. In addition, this type of normalization is also applied in the MEGAN model (Guenther et al., 2012) and hence allowed us to compare and fit our data to the MEGAN algorithm used to predict the $CO_2$ response of isoprene emissions. To describe the global responsiveness of a variable to low and high $CO_2$ of a given response curve, we used the mean values of the 400-$CO_2$-normalized data $X_{<400} X_{400}^{-1}$ (n=2) and $X_{>400}$

$X_{400}^{-1}$ (n=4) for the measurements below and above 400 $CO_2$, respectively. The relative change in Fv/Fm and dark respiration (R) was expressed as the difference between the value before and after $CO_2$-ramping divided by its initial value (e.g. δ Fv/Fm $= (Fv/Fm_{ini}-Fv/Fm_{end}) Fv/Fm_{ini}^{-1}$).

The influence of growth conditions were assessed using Analysis of Variance on each data set of the two assay temperatures after having tested for normality (Shapiro-Wilk test) and equal variance (Levene test). If tests failed, the non-parametric

Kruskal-Wallis test was applied. Post-hoc Tukey HSD and Dunn's tests were used for pairwise comparison. The influence of assay temperature on pooled data of growth conditions was examined using Student or Mann-Whitney rank sum test. Paired Student tests or Wilcoxon signed-rank tests were used to compare data of two assay $[CO_2]$ (400 vs 800 ppm) measured during $CO_2$-ramping on a same leaf. The differences between groups of measured variables were considered to be significant at the level α = 0.05. Pearson correlation analyses were performed in order to test the covariation among variables. Consistency of

correlations (linearity, outliers) was visually checked by scatter plots. In general, the data distributions seen on scatter plots provided no clear evidence for non-linear relationships with little exception: the relation between $G_{400}$ and $A_{400}$ was slightly





curved at high values ($A_{400}$ changed somewhat less with increasing $G_{400}$). All statistical analyses were done with addinsoft (2021) XLSTAT statistical and data analysis solution except the non-linear regression analyses (curve fitting), which was done with the SigmaStat 2.0 Jandel Scientific Software.

**3 Results**

**3.1 BVOC emission pattern and chemotypes**

Foliar VOC emission of all oak saplings were mainly composed of five MTs α-pinene, sabinene, β-pinene, myrcene and limonene accounting to 95 ± 5 %. The remainder was composed of α-tujene, camphene, 1.8-cineol and β-ocimene. Individual trees released the five major MTs in different proportions, roughly according two type of emission profiles: About two thirds 235 of the trees (17 of 26) produced α-, β-pinene and sabinene in high proportions (60-90 %), while one third (9 of 26) high proportions of limonene and myrcene (60-90 %). All replicate measurements made on a same or different leaf of individual trees showed that the relative proportions of these 5 VOCs were not different between leaves and not influenced by assay temperature or [$CO_2$], and hence tree specific (chemotype) (Fig. S2 in Supplement 1). Apart from the emission composition, there was no apparent difference between the two chemotypes in any of the measured variables including the total VOC 240 emission rate and responses to $CO_2$ (Figs. S3 and S4 in Supplement 1). These observations allowed us to restrict our data analyses on the sum of the major compounds.

**3.2 Intraspecific variability of the foliar emission factor**

The mean emission rates across all growth treatments of the sum of the 5 major emitted MTs measured at the beginning of the $CO_2$-ramping were 1491 ± 537 and 2456 ± 865 ng m$^{-2}$ s$^{-1}$ (11.0 ± 4.0 and 18.1 ± 6.4 nmol m$^{-2}$ s$^{-1}$, 32.6 ± 11.4 and 55.8 ± 21.4 245 µg g$^{-1}$ h$^{-1}$) for 30 °C and 35 °C assay temperature respectively. The deduced temperature-normalized EF ranged by a factor 4 between 610 and 2686 ng m$^{-2}$ s$^{-1}$ (4.5-19.7 nmol m$^{-2}$ s$^{-1}$, 13.1-60.8 µg g$^{-1}$ h$^{-1}$) and averaged on 1694 ± 589 ng m$^{-2}$ s$^{-1}$ (37.6 ± 13.6 µg g$^{-1}$ h$^{-1}$). There was no significant difference between the mean EF deduced from the 30°C and that from the 35 °C measurements (1626 ± 575 vs. 1781 ± 605 ng m$^{-2}$ s$^{-1}$ (P = 0.38, t-test on merged data of growth treatments). There was also no significant difference between the mean EFs or other variables of the four growth regimes, except a significant difference in 250 plant growth (ANOVA, P= 0.030) with the lowest value observed for the plants grown under the 400/25 regime (Table S2 in Supplement 1). Pooling the data of the two growth temperature regimes revealed that growth under elevated $CO_2$ significantly increased seedling growth (t-test, P=0.008) and LMA (t-test, P=0.022), though it had no effect on estimated foliar chlorophyll content (t-test, P=0.296). Growth at double $CO_2$ had also a general positive effect on the mean EF per leaf surface determined at 30° assay temperature (t-test, P=0.012; Table S2). This difference was however not significant for EF per leaf dry mass (t-255 test, P=0.087). The EFs deduced from the 35°C measurements were not different between the two growth [$CO_2$]. However, the initial C-loss at 35 °C ($E_{400}/A_{400}$) and the initial respiration rate ($R_{ini}$) were respectively lower and higher in leaves grown





under elevated $CO_2$ than under normal $CO_2$ (t-tests, P=0.029 and 0.015). Pooling the data of the two growth $CO_2$ regimes did not show any significant effect of growth temperature on plant growth, leaf structure (LMA, [Chloro]), EFs or any photosynthetic variable measured at the beginning of a series at either assay temperature.


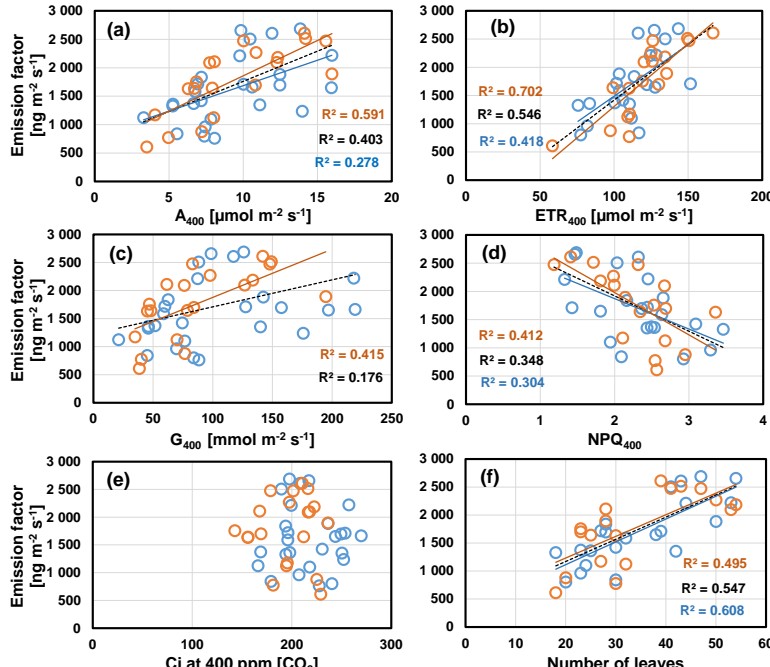

**Figure 1.** Scatter plots of the foliar emission factors measured at the beginning of $CO_2$-ramping (400 ppm $CO_2$) and assay temperatures of 30 °C (blue) and 35 °C (red) against simultaneously measured photosynthetic variables (a)-e)) and the number of leaves of the plants (f)). Lines with determination coefficients $R^2$ show best-fit results from Pearson correlation analyses with $P < 0.05$. Broken lines and $R^2$ in black

are from pooled data. See Table S3 in Supplement 1 for more information.

Overall, the results showed a large variability of EF within all growth regimes. Correlation analyses revealed that at 30 °C assay temperature the leaf-to-leaf variation in EF co-varied most strongly with the plants growth rate (Fig. 1 (f) and Table S3 in Supplement 2). In addition, EF was positively correlated with the leaf's actual assimilation ($A_{400}$) and electron transport rate

($ETR_{400}$) along with a weak negative correlation with $NPQ_{400}$ (Figs. 1(a), (b,) (d)). The same relationships held at 35 °C assay temperature. However, the correlation of the deduced EFs with the plant's growth rate was less strong than at 30 °C, while those with ongoing photosynthetic processes $A_{400}$, $ETR_{400}$ and $NPQ_{400}$ were strengthened including a positive correlation between EF and stomatal conductance $G_{400}$ (Fig. 1(c)). By contrast, at neither assay temperature, EF variability was correlated to that of the initial Ci, which ranged between 150 and 260 ppm (Fig. 1(e)).


### 3.3 Response pattern of VOC emissions and photosynthetic variables during $CO_2$-ramping

Control runs, in which repeated measurements were taken during the course of the day while temperature, PPFD, and [$CO_2$] were held at standard conditions showed that leaf emission rates changed very little during the day (Fig. S5 in Supplement 1), ruling out a possible major bias in detecting $CO_2$ effects due to endogenous diel variation in leaf emissions.

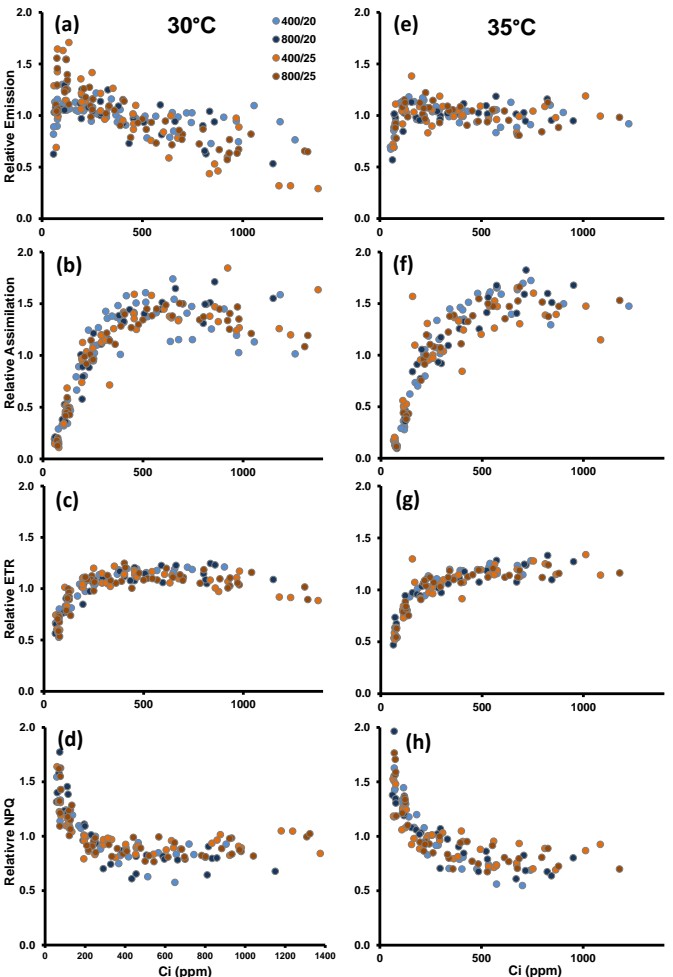

**Figure 2.** Relative MT emission, $CO_2$-assimilation, electron transport rate and non-photochemical quenching against calculated leaf internal $CO_2$ concentration (Ci) measured during $CO_2$ ramping at assay temperatures of 30 °C (left panels) and 35 °C (right panels). Altogether, 26 and 20 $CO_2$-response curves were run at 30 and 35 °C, respectively. Colors of the dots denote the temperature and $CO_2$ regimes, in which plants have been grown. Data were normalized by devising the individual data of a $CO_2$-ramping curve by its mean. To compare the overall amplitude of responses, all y-axis were set to the same scale.

The relative changes of MT emissions and photosynthetic variables during $CO_2$-ramping exhibited different pattern according to the assay temperature, regardless of whether the emission data were normalized to the means per series (Fig. 2) or to the





initial measurements at 400 ppm $CO_2$ (Fig. S6 in Supplement 1). Indeed, comparing the averages of $E_{400}$ normalized emissions across all populations, relative emission changes at both low or high $CO_2$ were significantly different between the two assay

temperatures (t-tests, $E_{<400}$ $E_{400}^{-1}$: P=0.021; $E_{>400}$ $E_{400}^{-1}$: P<0.001). At 30 °C (Fig. 2(a)-(d), Fig. S6(a)-(d)), emissions frequently decreased under high $CO_2$ ([$CO_2$] > 400) and showed variable responses to low $CO_2$ ([$CO_2$] < 400) including increases and decreases. Photosynthesis continuously increased until 400 to 600 ppm Ci, leveled off beyond with occasional decreases at highest Ci. The amplitude of change in ETR was smaller than in photosynthesis. Nevertheless, it generally dropped at Ci lower than 200 ppm and tended to decrease at highest Ci. The pattern of NPQ changes somewhat mirrored that of ETR. NPQ

increased at Ci below 400 ppm and mostly remained unchanged or slightly decreased at higher Ci. Fv/Fm values were significantly lowered after $CO_2$-response curves (p <0.001) indicating that leaves did not fully recover from photoinhibition that occurred during $CO_2$-ramping ($\delta$Fv/Fm, Table S1 in Supplement 1). At 35 °C assay temperature (Fig. 2(e)-(h), Fig. S6(e)-(h) in Supplement 1), the relative emission rates expressed a less variable responsiveness to $CO_2$ than at 30 °C. It more frequently decreased at low Ci than at 30 °C but remained largely insensitive to high Ci. Photosynthesis leveled off later at

higher Ci than at 30°C with no or less inhibition at highest Ci. Similarly, ETR never decreased during high $CO_2$ exposure compared to 30 °C but rather slightly increased with increasing Ci. Analogously, during the ramping to high $CO_2$, relative NPQ decreased more and leveled off later at 35 °C than at 30 °C. The decrease in Fv/Fm was lower at 35 °C than at 30 °C, though the difference was not significant ($\delta$Fv/Fm: 8 ± 4% *vs.* 11 ± 4%, t-test: P=0.06).

Correlation analyses (Table S3 in Supplement 2) revealed that at both assay temperatures, mean relative emissions at low $CO_2$

($E_{<400}$ $E_{400}^{-1}$) scaled positively with those of ETR (P = 0.004; Fig. 3(a)) and negatively with the leaf's initial C-losses ($E_{400}/A_{400}$) measured at the beginning at normal [$CO_2$] (30 °C: R = -0.52, P = 0.006; 35 °C: R = -0.51, P = 0.021; data not shown). However at 35 °C, $E_{<400}$ $E_{400}^{-1}$ was also strongly correlated with the leaf's initial photosynthesis $A_{400}$ (P = 0.001; Fig. 3(b)) and stomatal conductance rate $G_{400}$ (P < 0.001; Fig. 3(c)). These correlations were not significant at 30 °C, mainly because two leaves exhibited increased emissions at reduced [$CO_2$] along with a relatively high ETR, while their initial photosynthetic and stomatal

conductance rates were rather low. At either assay temperature, $ETR_{<400}$ $ETR_{400}^{-1}$ was unrelated to $A_{400}$, $G_{400}$ and C-loss$_{400}$. During subsequent ramping to high $CO_2$, the emission reductions observed at 30 °C ($E_{>400}$ $E_{400}^{-1}$) were best explained by concomitant reductions in ETR (P < 0.001; Fig. 3(d)) and, anti-correlated with ETR, by increases in NPQ (P = 0.003; Fig. 3(e)). $E_{>400}$ $E_{400}^{-1}$ was not related to any other variable except a weak negative correlation with the relative emissions before to low $CO_2$ $E_{<400}$ $E_{400}^{-1}$ (R = -0.44, P=0,026, data not shown). By contrast, the small variable emission changes under high $CO_2$

at 35 °C showed a strong positive correlation with the relative emission rates measured before under low $CO_2$ (R = 0.71, P<0,001). This correlation should be considered with caution. Due to their interdependency (common denominator), random variation in the absolute emission rates associated with limited precision in BVOC measurements generate positive correlations without any $CO_2$ effect (Fig. S1 in supplement 2). Finally when data of both assay temperatures were pooled, relative emissions at high $CO_2$ $E_{>400}$ $E_{400}^{-1}$ data were negatively correlated with Fv/Fm reductions ($\delta$Fv/Fm: P=0.014; Fig. 3(f)). Also of note,

Open Access EGU

δFv/Fm was negatively correlated with plant growth at both assay temperatures (30 °C: R = -0.51, P = 0.015; 35°C: R = -0.67, P=0.003).

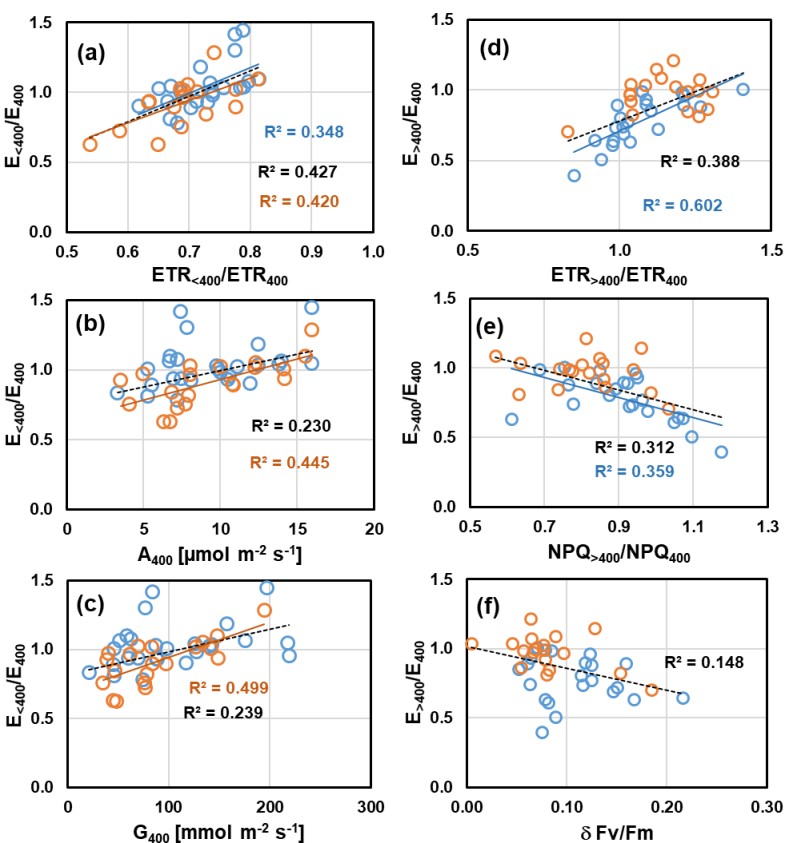

**Figure 3.** Key correlations of relative emissions to low (left panels) and high $CO_2$ (right panels) measured at assay temperatures of 30 °C (blue) and 35 °C (red). Relative rates of a measured variable are calculated as the mean of $V_{<400}/V_{400}$ and $V_{>400}/V_{400}$ where $V_{400}$ is the value

measured at the beginning of $CO_2$-ramping at 400 ppm $CO_2$, and $V_{<400}$ and $V_{>400}$ are the values subsequently measured at lower and higher [$CO_2$]. Lines with determination coefficients $R^2$ show best-fit results from Pearson correlation analyses with P <0.05. Lines and $R^2$ given in black are from pooled data.

$CO_2$-responses of emissions and photosynthetic variables did not differ significantly among the four growth populations with

one exception (Table S2 Supplement 1): At 30 °C, there was a significant difference in relative NPQ at high $CO_2$ ($NPQ_{>400}$ $NPQ_{400}^{-1}$) with 800/25 and 400/25 grown plants showing less reduction in NPQ than 800/20 grown plants (ANOVA, P = 0.006). When the data of the two $CO_2$ growth regimes were pooled, growth temperature significantly affected the high-$CO_2$-responses of emissions, ETR and NPQ at 30 °C ($E_{>400}$ $E_{400}^{-1}$: t-test, P=0.008; $ETR_{>400}$ $ETR_{400}^{-1}$: t-test, P=0.005; $NPQ_{>400}$ $NPQ_{400}^{-1}$: t-test, P=0.001). In fact, warm grown plants mostly continued to non-photochemically dissipate light energy to the

expense of ETR and MT emissions, whereas NPQ of cool grown plants frequently relaxed during high-$CO_2$-ramping along with keeping higher ETR and emission rates. Accordingly, Fv/Fm was significantly more reduced in warm grown plants



compared to cool grown plants (δFv/Fm: t-test, P=0.016). Growth temperature also affected the $CO_2$-responsiveness of some photosynthetic variables at 35 °C assay temperature (table S2 Supplement 1): Leaves grown under elevated temperature opened less at low $CO_2$ and closed more stomata at high $CO_2$ than in leaves grown under low temperature (t-tests: $G_{<400}$ $G_{400}^{-1}$: P=0.009,

$G_{>400}$ $G_{400}^{-1}$ : P=0.003). Furthermore at high $CO_2$, warm grown leaves had lower $CO_2$-assimilation rates ($A_{>400}$ $A_{400}^{-1}$ : P=0.017) and higher NPQ ($NPQ_{>400}$ $NPQ_{400}^{-1}$ t-test, P=0.001) and ETR/A ratios ($ETR/A_{>400}$ $ETR/A_{400}^{-1}$ , t-test, P=0.014) compared to leaves grown under low temperature. However, growth temperature had no significant effect on emission responses to low and high $CO_2$ at 35 °C. Pooling the data of the two growth temperature regimes did not reveal any effect of growth $CO_2$ on $CO_2$-responsiveness of emissions or photosynthetic processes.

The high-$CO_2$-inhibition of emissions at 30 °C can be simulated with the algorithm used in the MEGAN modelling framework (Guenther et al., 2012). Using the whole data set for the fit resulted in a response curve with coefficients close to that currently applied to predict the $CO_2$-responsiveness of isoprene emissions under current $CO_2$-level (Fig. 4, Table S4 in Supplement 1). However, as indicated by the statistics (see above) and can be seen from Fig. 4, relative emissions considerably varied with data from the low and high temperature grown plants predominately scattering above and below the total fit, respectively.

Consequently, separate fits resulted in two distinct curves, which differed mainly in the coefficient determining the amplitude of the emission reduction (C*).

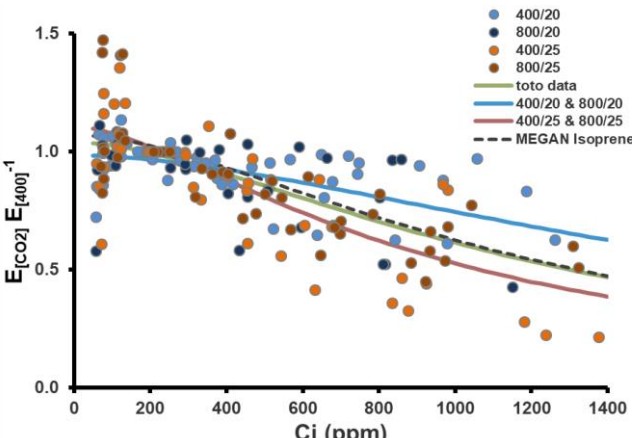

**Figure 4.** Relative MT emission rates (normalized to the initial measurement at 400 ppm $CO_2$) measured during $CO_2$-ramping at the assay

temperatures of 30 °C (26 response curves). Colors of the dots denote the $CO_2$ and temperature regimes, in which plants have been grown. Solid lines present best fits to the algorithm of the MEGAN modelling framework accounting for the short-term effect of $CO_2$ on isoprene emissions:  $C_{Ci} = E_{max} - (( E_{max} C_i^h ) / ( C^{* h} + C_i^h ))$, where $C_{Ci}$ is the $CO_2$ scaling factor, $C_i$ the leaf internal $CO_2$ concentration, and $E_{max}$, $C^*$ and $h$ are empirical coefficients. The algorithm simulates an inverse sigmoidal relationship between emissions and $C_i$, where $C_{ci}$ scales the emission rate at standard [$CO_2$] (400 ppm) to the progressive inhibitory effects of increasing $C_i$. The green line shows the fit from all

data and the red and blue lines from the warm and cool grown plants, respectively. Black broken line depicts the $CO_2$-scaling currently used in MEGAN (Guenther et al, 2012). All coefficients values and additional information are given in Table S4 of supplement 1.





To summarize, differences in the high $CO_2$-responsiveness of emissions were modulated by both actual leaf and the plant's growth temperature, albeit in opposite manner. The observed correlations suggest that the emission response to low $CO_2$
depended on the instantaneous energy status and the leaf's initial carbon balance, whereas the response to high $CO_2$ depended mostly on the instantaneous energy status, which was affected by photooxidative stress. Figure 5 provides a schematic overview on the key correlations regarding the initial emission factor and the relative emissions under low and high [$CO_2$].

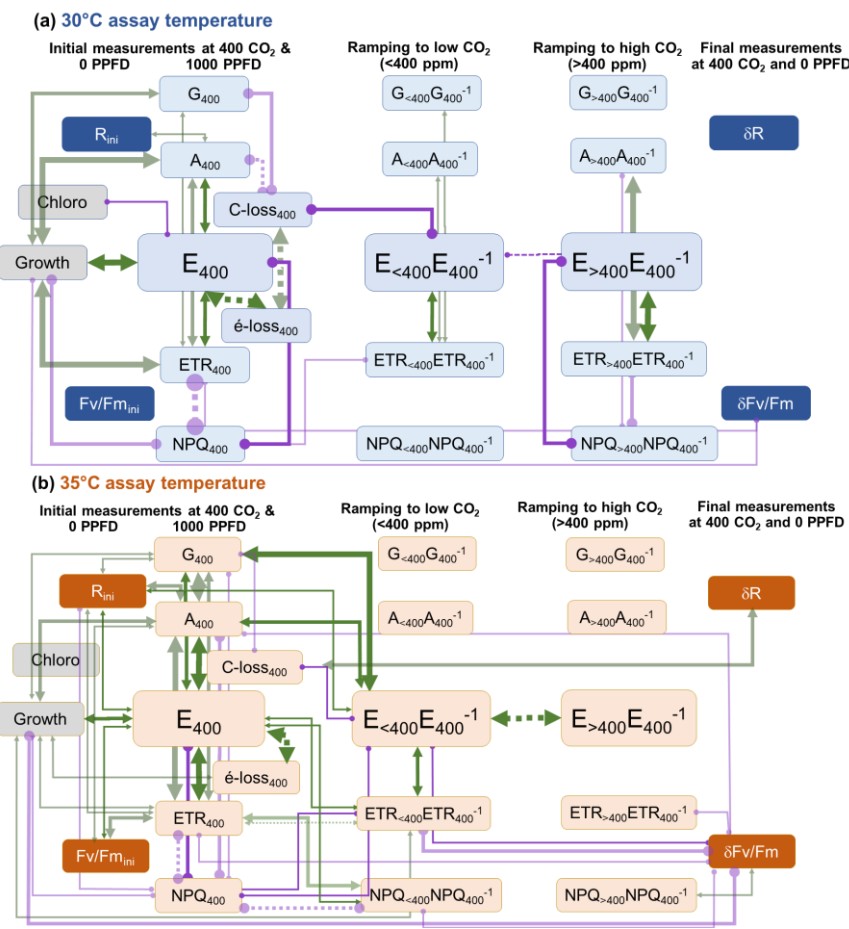

**Figure 5.** Overview of the correlation network between MT emission rates (E) and other variables measured during $CO_2$-ramping
experiments at two assay temperatures: 30 °C (a), 35 °C (b). Connectors represent significant positive (green) or negative (purple) relationships based on Pearson correlation analyses. Line thickness designates the level of significance of the correlation (thick: P <0.001, medium: $0.001 < P \leq 0.01$, thin: $0.01 < P < 0.05$). Not all significant correlations are shown; only those between emissions ($E_{400}$, $E_{<400} E_{400}^{-1}$, $E_{>400} E_{400}^{-1}$) and other variables (connectors in dark colours) plus the main correlations of the latter to third variables (connectors in bright colours) (see Table S3 in supplement 1 for a complete compilation). Correlations shown in broken lines should be interpreted with caution
since their variables are not independent (derived from common precursors). The results can be summarized as follows: The initial leaf emission factor measured at normal $CO_2$ ($E_{400}$) was positively related to plant growth performance (Growth) and current leaf photosynthetic processes of both light reactions ('energy status', $ETR_{400}$ and $NPQ_{400}$) and dark reactions ('carbon status', $A_{400}$, $G_{400}$). All these variables were



interrelated, but the apparent dependence of emissions from actual photosynthetic processes was stronger at 35 °C, as MT production was about twice as high as at 30 °C. During subsequent exposure to low $[CO_2]$, relative emission rates ($E_{<400} E_{400}^{-1}$) correlated with concurrent

changes in the leaf's energy status ($ETR_{<400} ETR_{400}^{-1}$) and, unrelated to this, with the initial carbon status ($A_{400}$, $C-loss_{400}$, $G_{400}$), the influence of the latter again being stronger at 35 °C than at 30 °C. During exposure to high $[CO_2]$, the emission reductions observed at 30 °C ($E_{>400} E_{400}^{-1}$) were principally correlated with the parallel evolution of leaf energy status ($ETR_{>400} ETR_{400}^{-1}$, $NPQ_{>400} NPQ_{400}^{-1}$). By contrast, emissions in response to high $[CO_2]$ at 35 °C changed little and showed only a pseudo-correlation with relative emissions at low $[CO_2]$, explained by their mathematical interdependency (see Fig. S1 in Supplement 2).

**4 Discussion**

**4.1 Variation in the foliar emission capacity - what makes the difference?**

The foliar emission factor was little affected by growth conditions, apart from the positive effect of elevated growth $CO_2$ on plant growth and LMA confirming a previous greenhouse study on QI (Staudt et al., 2001). On the other hand, the lack of a significant effect of growth temperature on EF observed in the present study contrasts to the results of a growth chamber

(Staudt et al., 2003) and several field studies (Peñuelas and Llusià, 1999; Staudt et al., 2002; Ciccioli et al., 2003; Lavoir et al., 2009) suggesting that the EF of QI leaves undergoes strong seasonal cycles related to prevailing meteorological conditions. We explain this contradiction as indicating that the long-term regulation of the leaf's emission capacity by temperature is non-linear and that persistent lower temperatures than that applied here are necessary for its down regulation, presumably associated with the expression and turnover of MT synthases (Fischbach et al., 2002; Lavoir et al., 2009). Furthermore, it is conceivable

that changes in the temperature regimes rather than constant temperature differences trigger acclimation processes in EF (Wiberley et al., 2008). Finally, moderate effects of growth temperature on EF may have been overlooked due to its large variability within the four populations.

EF variability scaled positively with the leaf's actual photosynthetic activity and the individual plant's growth rate, which was negatively related to the subsequent loss in Fv/Fm during $CO_2$-ramping. This suggests that the overall sink capacity of

individual plants was associated with the capacity of its leaves to fix carbon and produce MTs and to avoid persistent photoinhibition. Positive relations between isoprenoid emissions, photosynthesis, growth performance and resistance to harsh environmental conditions within and across natural or genetically manipulated populations have already reported (e.g. Monson and Fall, 1989; Staudt et al., 2001; Possell et al., 2004; Eller et al., 2012; Lantz et al., 2019a; Zuo et al. 2019; Niinemets et al., 2021; Dani et al. 2022), though also opposite or no relations have been observed (e.g., Guidolotti et al., 2011; Behnke et al.,

2012; Zuo et al., 2019; Monson et al., 2020). The positive association we observed on QI saplings may due to the beneficial effects of chloroplastic volatile isoprenoid production on their photosynthetic and growth performances by modulating cellular signaling networks optimizing plant growth and stress resistance (Frank et al., 2021; Monson et al., 2021; Dani et al., 2022). However, because photosynthetic processes supply directly chloroplastic isoprenoid biosynthesis with carbon substrates and energetic cofactors, it is also conceivable that oppositely higher foliar emission capacity of QI saplings resulted from their



higher foliar photosynthetic activity constrained by the plant's sink capacity to use photosynthates (Ainsworth and Bush, 2011).
       Covariation of leaf emission and photosynthesis might also reflect variation in leaf anatomy as for example the per leaf surface
       quantity of photosynthetically active tissues and density of chloroplasts ((Sun et al., 2012; Rasulov et al., 2015). We found no
       consistent correlation between estimated chlorophyll content or LMA and initial photosynthesis or emission rates not
       supporting leaf anatomical differences being the major cause of the observed correlation pattern. Instead, a portion of

variability in initial $A_{400}$ was simply related to leaf-to-leaf variability in stomatal conductance $G_{400}$. Especially at 35°C assay
       temperature, when terpene production was almost double as high as at 30°C, EF scaled positively with $G_{400}$ along with $A_{400}$
       and $ETR_{400}$, while at 30°C it was more related to the plant's growth performance (Fig. 5). Considering that stomatal opening
       does not directly control MT emissions (Niinemets et al., 2014), this shift in the correlation pattern indicates that at 35 °C the
       chloroplastic MT production was frequently limited by ongoing $CO_2$-assimilation via the supply of basic carbon substrates and

energetic cofactors. Instead at 30°C, the leaf-to-leaf variability in the more moderate MT production might have more
       frequently mirrored the variability in the foliar amount of active MT synthases (see e.g. Loreto et al., 2001; Fischbach et al.,
       2002) and/or other rate-limiting enzymes of the MEP pathway (Lantz et al., 2019a), which seemingly was associated with the
       plant's growth performance perhaps due to a congruent variability in growth hormones produced within the same metabolic
       pathway (Dani et al., 2022). By contrast, $CO_2$-responsiveness was not the reason for the leaf-to-leaf variability of EF (for an

example see Guidolotti et al., 2011), since it was unrelated to initial Ci at both assay temperatures (Fig. 1(e)).

**4.2 What are the drivers of the $CO_2$-responsiveness?**

       If leaf MT biosynthesis was constrained by low G and A already at the beginning at 400 ppm $CO_2$, it is likely that emissions
       decline under low $CO_2$-ramping when photosynthetic processes became rapidly reduced. Indeed at 35°C all leaves having
       generally low initial assimilation and stomatal conductance rates showed the most pronounced emission decrease under low

$CO_2$, whereas at 30°C only leaves having low assimilation together with high emission rates (hence high C-losses). In addition
       and unrelated to the leaf's initial photosynthetic and MT production status, relative emission rates under low $CO_2$ ($E_{<400}$ $E_{400}^{-1}$) scaled positively with relative ETR ($ETR_{<400}$ $ETR_{400}^{-1}$) at both assay temperatures. We interpret these observations that MT
       production during low $CO_2$-ramping was curbed by two rather independent constraints, one associated with the availability of
       basic C3-carbon substrates entering in the MEP pathway and one with the availability of energetic co-factors necessary to

reduce them further downstream. The latter predominated initially at moderate low $CO_2$ when energetic cofactors were still
       primarily used in the CBB-cycle for $CO_2$-reduction and photorespiration, whereas the former when Ci approached the $CO_2$
       compensation point (i.e., when A = 0) and the more and earlier the leaf initial emission rates were high and assimilation low.
       Labelling studies have shown that the fraction of 'older' carbon incorporated in the biosynthesis of isoprene increases during
       exposure to low [$CO_2$] and/or high temperatures (Funk et al., 2004; Trowbridge et al., 2012; de Souza et al., 2018; Guidolotti

et al., 2019; Yanez-Serrano et al., 2019) as well as leaf-internally recycled respired $CO_2$ (Garcia et al., 2019).
       Relative ETR was also by far the best predictor of emission changes to high $CO_2$ at 30°C suggesting that the same mechanisms
       contributed to modulate emissions at moderate low and at high $CO_2$. Earlier studies on isoprene emissions suggested that this





high-$CO_2$ inhibition results from an activation of the cytosolic PEP-carboxylase under high [$CO_2$] (but see Abadie and Tcherkez, 2019) leading to a reduction of PEP available for import into chloroplasts and in turn less pyruvate for isoprenoid

biosynthesis (Rosenstiel et al., 2003). This hypothesis however does not explain its temperature dependency (Sun et al., 2013; Monson et al., 2016) and was not confirmed by studies using competitive PEP-carboxylase inhibitors (Rasulov et al., 2018). An alternative hypothesis links the emission reduction at high $CO_2$ to the occurrence of feed-back inhibition of photosynthetic processes (Sharkey and Monson, 2014; and references therein). Lack of increase or decreases of A and ETR at high $CO_2$ are typically observed when the production of triose phosphate in the CBB-cycle exceeds its utilization for starch and sucrose

synthesis (TPU-limitation; McClain and Sharkey, 2019). The accumulation of sugar phosphates will lead to the depletion of inorganic phosphate (Pi) necessary to sustain ATP synthesis and ultimately inhibiting photosynthetic light and dark reactions, and also the availability of pyruvate inside the chloroplasts by compromising the exchange rates to the cytosol via Pi transporters (Sharkey and Monson, 2014; de Souza et al., 2018). TPU-limitation of photosynthesis occurs less under high temperature, mostly because sucrose synthesis is enhanced thereby restoring Pi levels. Beside its direct effect on metabolic

rates, high temperature decreases $CO_2$-solubility (in pure water approx. -10 % from 30° to 35 °C) possibly lowering photosynthetic reduction of $CO_2$ coming from both external (ambient air) and internal (respiration) sources (Potosnak et al., 2014). Growth conditions can affect TPU. Plants acclimatized to low temperature or elevated $CO_2$ tend to have an increased Pi regeneration capacity thus being less vulnerable for TPU limitation (McClain and Sharkey, 2019). Indeed in this study, signs of TPU-limitation were virtually absent at 35 °C assay temperature, and at 30 °C, the strongest decline in emission and

photosynthetic processes was observed on 400/25 grown plants showing the lowest growth, indicating that a limited capacity to use photosynthates for growth favored TPU-limitation of photosynthesis and inhibition of emissions. On the other hand, there was no overall correlation between the plant's growth rates and their emission responsiveness to high $CO_2$ (Table S3 Supplement 2, Fig. 5). Furthermore, comparing the shape of response curves of individual normalized E and ETR data rather than their means, suggests that their evolutions were partly disconnected in opposite ways during high and low $CO_2$-ramping

(Fig. S6 in Supplement 1): During low $CO_2$-ramping, emissions increased occasionally while ETR always dropped, and during high $CO_2$-ramping, emissions frequently decreased earlier and more than ETR. In fact, ETR rarely decreased at Ci lower than 800, which is consistent with the global median Ci value for TPU-limitations deduced by Kumarathunge et al. (2019). Similarly, Monson et al. (2016) and Lantz et al. (2019b) reported that isoprene emissions decreased in response to high $CO_2$ before TPU-limitation appeared. Thus, unless TPU limitations occurred before being detectable by gas exchange and

fluorescence measurements (Sharkey, 2019), MT emission started to decrease when ETR was mostly insensitive to increasing $CO_2$, i.e. when the CBB-cycle is typically limited by the production of energetic cofactors from ETR to regenerate the primary $CO_2/O_2$-acceptor Ribulose-1,5-bisphosphate. At this stage, A is expected to slightly increase with increasing Ci, because photorespiration, the second most important electron sink, is progressively inhibited, as evidenced by a decrease in the ETR/A ratio. As a result, energetic cofactors might be less available for the MEP pathway, notably reduction power, which is consumed

more during $CO_2$-reduction than during photorespiration with respect to ATP (Niinemets et al., 2021; and references therein).This hypothesis has been used as the basis for photosynthesis-linked modelling of isoprene emissions ('excess energy



model', e.g. Morfopoulos et al., 2014). In the same context, it was suggested that the emissions of isoprene and MTs were positively related to NPQ, indicating an excess of reducing power for the biosynthesis of isoprenoids (Peñuelas et al., 2013; Filella et al., 2018). The lack of high-$CO_2$-inhibition of emissions at higher temperatures is explained by the fact that a higher

ETR is usually maintained without being consumed in the CBB-cycle. Our results provide partial support for the excess energy hypothesis: The ETR/A ratios were indeed generally higher at 35 °C than at 30 °C due to higher ETR at 35 °C (Table S2 in Supplement 1). The difference was only significant for warm grown plants consistent with their stronger emission responsiveness to high $CO_2$. Furthermore, during initial $CO_2$-ramping to low $CO_2$ the ETR/A ratios strongly increased (Fig. S7(a) in Supplement 1), which would explain why emissions sometimes increased under moderate low $CO_2$ at 30 °C before

presumably being constrained by the lack of basic C3-intermediates (see discussion above). Yet, in response to high $CO_2$, ETR/A-ratios decreased little, and the variations were less related to the emission changes than ETR (Table S3 in supplement 2 and Fig. S7(b), (c) in Supplement 1). Also, our study showed not a positive but a clear negative correlation between NPQ and emission responses to high $CO_2$ indicating that the maintenance of NPQ processes during high $CO_2$ co-constrained ETR, A and MT biosynthesis. Given that our plants were adapted to greenhouse light conditions, the continuous exposure to a

relatively high PPFD level at various [$CO_2$] caused some persistent photoinhibition as evidenced by the loss of Fv/Fm. Especially during exposure to lowest [$CO_2$] when ETR and A rapidly declined, electrons or excitation energy were likely transferred to $O_2$ generating ROS. NPQ and ROS formation efficiently reduces the availability of reduction power in two ways: First by reducing its formation during PSII electron transport by diverting the absorption or the absorbed light energy from PS (thus lowering ETR), and second, by enhancing its consumption for ROS detoxification and NPQ mechanisms inside

chloroplasts notably in redox reactions associated with the xanthophyll cycle, the water-water cycle starting with the Mehler reaction, the glutathione-ascorbate cycle and the ferredoxin thioredoxin system (for overviews see e.g. Asada, 2006; Foyer and Noctor, 2011; Choudhury et al., 2016, Ruban, 2016; Kang et al. 2019). We speculate that during initial high $CO_2$-ramping, when ETR and A frequently co-evolved and were still not feedback inhibited, a portion of PSII electrons was diverted from MT-biosynthesis and the CBB cycle for repair and protective mechanisms. This initial decrease depending on NPQ relaxation

was essentially caused by the lack of reduction equivalents while ATP synthesis was still maintained by linear and cyclic electron flow, which subsequently became rapidly compromised with the onset of TPU limitation thus explaining the uneven, non-linear character of high-$CO_2$ inhibition. We are unable to quantify the losses of photochemical energy and carbon precursors linked to the photooxidative stress that occurred in our experiments. Based on the measured Fv/Fm values, the total loss of the leaf's capacity for PSII electron transport during $CO_2$ ramping was about 10%, to which would be added the

reduction in availability of reduction equivalents due to their use in photostress-related redox systems. For comparison, the calculated amount of electrons spent for MT emissions (é-losses) rarely exceeded 1 % ($0.57 \pm 0.17$ and $0.80 \pm 0.21$ at 30 °C and 35 °C respectively), of which less than the half is used in the MEP-pathway (12 of the total 28 moles reduction equivalents for 1 mole MT; Sharkey and Monson (2014)). Hence, the fraction of excess electrons used for MT synthesis was very small compared to the total stress-related reduction and alternative electron sinks in general (Dani et al., 2014b). Furthermore,

photooxidative stress also occurred at 35 °C (albeit to a lower extent than at 30 °C), when emissions were much higher but





hardly affected by high-$CO_2$ exposure. These facts collectively shed doubts that the loss of ETR by photooxidative stress was the sole cause of the initial high-$CO_2$ emission inhibition.

In summary, while the data of our study clearly indicate a link between photosynthetic electron transport and $CO_2$-responsiveness of MT emissions from QI leaves, they do not allow us to confirm or refute current hypotheses about the $CO_2$
sensitivity of isoprenoid emissions, which might not be mutually exclusive but have interacted during the course of $CO_2$-ramping. Shifts in the availability of limiting metabolites are complex and possibly did not really reach a steady state within the applied time steps of our protocol, but continuously adjusted and fluctuated among sources and sinks of energetic cofactors constantly interacting with the availability of carbon intermediates including feedback and feedforward controls within the MEP-pathway (for overview see Sharkey and Monson, 2014; and de Souza et al., 2018). At timescales over hours, theses shifts
have likely included regulations of enzymes activities at transcriptional level (e.g. Hartikainen et al., 2012; Kanagendran et al. 2018). Photooxidative stress for example induces the biosynthesis of downstream higher isoprenoids such as carotenoids and tocopherols, possibly via retrograde signals of ROS, MEP-pathway precursors or carotenoid degradation products (Xiao et al. 2012; Ramel et al., 2013; Foyer, 2018; Jiang and Dehesh, 2021), which in turn could curtail the production of MTs through competition for the same precursors, or oppositely enhancing it by relieving feedback inhibitions in the MEP-pathway and
keeping a higher level of MT precursors (Behnke et al. 2009; Banerjee et al., 2013; Ghirardo et al., 2014; Rasulov et al., 2014; Zuo et al. 2019; Sun et al., 2020 and references therein).

## 4.3 Relevance of results for predicting MT emissions in a future warmer high-$CO_2$ world

Generally, the extrapolation of the emission responses to high $CO_2$ we observed should be viewed with caution, since plants under natural conditions do not undergo $CO_2$-response curves like those in our study, in which emission reductions were
partially related to simultaneous inhibition of photosynthetic processes and oxidative stress. On the other hand, we found little evidence that emission inhibition by high $CO_2$ was influenced by the preceding exposure to low $CO_2$. Furthermore, the occurrence of oxidative stress associated with photosynthetic feedback limitation is more likely in a high $CO_2$ world. Keeping this in mind, we attempted to estimate the extent to which the observed emission inhibition by double [$CO_2$] in a future warmer world could offset the emission increase by the congruent increase in temperature. Doubling assay [$CO_2$] from 400 to 800 ppm
reduced temperature normalized emission rates on average by 8 % at 30 °C ($1626 \pm 575$ vs. $1499 \pm 538$; P = 0.001; paired t-test; n = 26) and had no effect at 35 °C ($1781 \pm 605$ vs. $1800 \pm 719$; P = 0.755; paired t-test; n = 20). Regarding the individual growth populations, double [$CO_2$] significantly decreased 30°-emissions only in the 800/25 and 400/20 grown plants by respectively 10 % and 5 %. Yet, it is very likely that the high-$CO_2$ inhibition becomes stronger at temperatures lower than 30°C. Several studies on isoprene emissions reported a negative linear relationship between high-$CO_2$ inhibition and leaf
temperature perhaps dumping emissions at 800 ppm $CO_2$ by 40 % or more under lowest temperatures (Rasulov et al., 2010; Sharkey and Monson, 2014; Zuo et al., 2019; Niinemets et al., 2021; and references therein). In turn, the short-term increase in emissions per one degree temperature increase is about 13 % in the range between 10 and 30 °C and then gradually decreases until the temperature optimum, which is around 41 °C for QI emissions (Staudt and Bertin, 1998). In addition, the EF of the





evergreen QI foliage strongly diminishes towards the winter season (Peñuelas and Llusià, 1999; Staudt et al., 2002; Ciccioli
et al., 2003) along with the down-regulation of the activity of MT synthases (Fischbach et al., 2002; Grote et al., 2006). Both,
seasonal and diel courses of QI emissions typically resemble a broad dome-shaped peak. Since high-$CO_2$ inhibition would be
particularly relevant during the cooler daytime hours and all along the cooler seasons, diel and seasonal courses of QI emissions
should become higher but narrower in a future warmer high-$CO_2$ world with respect to the today emission courses. The $CO_2$-
effect on the total annual MT release by a leaf might be limited as long as the warmer climate boosts summer emissions while
reducing its high-$CO_2$ inhibition. To assess the relative importance of the observed $CO_2$ inhibition, we computed annual
emissions by combining eight high-$CO_2$-inhibition scenarios differing in their maximum high-$CO_2$ inhibition with four
warming scenarios (1-4 °C warming) and three scenarios of EF seasonality (seasonality without and with summer drought, no
seasonality). All $CO_2$ inhibition scenarios assume that emission inhibition is zero at an air temperature of 35 °C or higher and
increases by 2 % per 1°C-decrease to reach eight different maximum inhibitions varying between 10% and 100%. Hence in
all scenarios the inhibition is the same between 30° C (10% as observed on elevated $CO_2$ and warm-grown plants) and higher
temperatures. Simulations were based on 30-min climate data recorded over three years (2019-2021, annual mean temperature:
14.5 ± 0.4) by a flux tower in a nearby QI forest station, EF seasonality of a former rain exclusion experiment on QI trees near
the institute (Staudt et al., 2002) and the light and temperature algorithms described in Staudt and Bertin (1998). On the whole,
96 simulations were run, whose outputs are summarized in Fig. 6 and Table S5 in Supplement 1 (see also Fig. S7 for an
example of the diurnal and annual emission courses resulting from simulations). As expected, the relative overestimation of
the future annual emissions when neglecting high-$CO_2$ inhibition will decrease with the level of climate warming (Fig. 6(b)).
Depending on the assumed maximum $CO_2$-inhibition, it ranges between 8 and 15 % for the 1°-warming and between 6 and 11
% for the 4°-warming scenario. Emission inhibition by double $CO_2$ would compensate or decrease the enhancement of annual
foliar emissions by warming only if global warming stays below 1.5 °C (Fig. 6(a)), which is rather unlikely under double [$CO_2$]
(IPCC 2021). Thus, although $CO_2$-inhibition of emissions will be close to or even at its maximum level during most time of
the year, its impact on the annual emission budget will remain modest due to the instantaneous non-linear effects of temperature
and light on emissions, and due to the strong seasonality of EF. In fact, during the winter season, the daily amounts of MTs
emitted by a leaf are about two orders of magnitude lower than those during the hot summer season (see examples in Fig. S7(e)
and (g) in Supplement 1). As a result, assuming higher maximum inhibitions than 40 % at temperatures below 15 °C has no
additional effect on the annual VOC budget (Fig. 6(b)). However, summer drought, which is likely to increase in intensity and
frequency in the future (Gao and Giorgi, 2008), can significantly curb emission seasonality (Staudt et al., 2002; Lavoir et al.,
2009), potentially enhancing the annual impact of $CO_2$-inhibition. Indeed, running simulations with EF seasonality from
drought-exposed QI trees instead of irrigated trees increased the annual $CO_2$-inhibition by 0.1-0.5 % (Table S5), which is
attributable to the lower annual EF maximum and its delayed occurrence in the summer season when decreasing day length,
light intensity, and temperature already constrain the daily VOC release. Assuming a complete suppression of EF seasonality
(constant mean EF throughout the year), increases annual $CO_2$-inhibition by 0.5-4 % (Table S5). On the other hand, our
simulations do not account for the effects of drought on Ci and leaf temperature, which can decrease and increase, respectively,
because of stomata closure, thereby reducing the high-$CO_2$ inhibition of emissions (e.g., Pegoraro et al., 2007). They also
ignore the potential positive effects of elevated $CO_2$ and warming on leaf phenology, longevity and the related duration of
emissions (Staudt et al., 2017; Mochizuki et al., 2020).

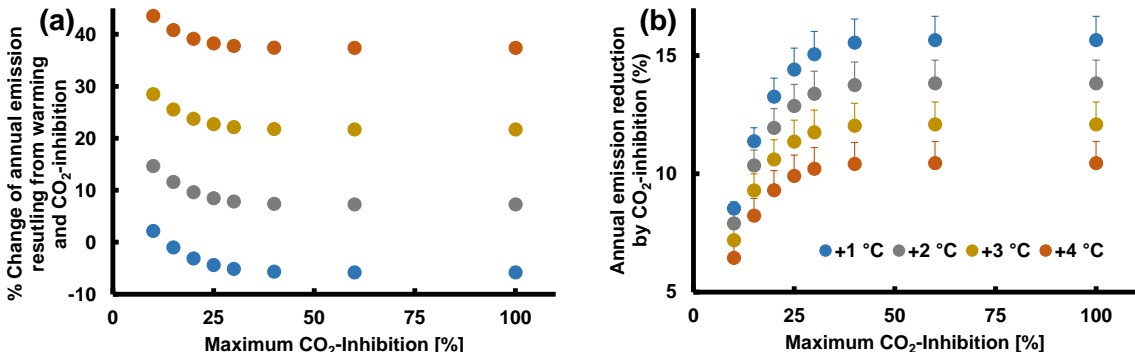

**Figure 6.** Assessment of the potential effect of high-$CO_2$ inhibition on total annual MT emission from QI leaves under a future warmer
climate with double [$CO_2$] combining different warming and $CO_2$ inhibition scenarios. To account for the high-$CO_2$ inhibition of emissions
at double $CO_2$, we assumed that inhibition is 0 % at ≥ 35 °C and progressively increases at lower temperatures by 2 % per 1 °C decrease to
reach maximum inhibitions of either 10, 15, 20, 25, 30, 40 of 100 %. For example for maximum inhibitions of 25 % and 40 %, emissions
become reduced by 25 % at ≤ 22.5 °C, and by 40 % at ≤ 15 °C, respectively. Data in (a) show the percentage change of annual emission
with respect to today values ((future annual $E_{+T+CO2}$ - today annual $E$) * today annual $E^{-1}$ *100), and (b) the percentage overestimation of
future annual emissions if $CO_2$-inhibition is ignored ((annual $E_{+T}$ - annual $E_{+T+CO2}$) * annual $E_{+T}$ $^{-1}$ *100). The simulations were based on the
temperature and PPFD data of the years 2019 to 2021 recorded on the flux tower located in a QI forest 25 km north-west of Montpellier
(annual mean temperature 14.5 ± 0.4 °C) and emission factors from Staudt et al. (2002) measured on non-water stressed adult QI trees in a
wood nearby the institute. Colours denote the different applied warming scenarios and error bars standard errors of the means of n=3 years
(not visible in (a)). For further examples, additional explanations and illustration of resulting annual and diel emission courses see Table S5
and Fig. S6 in Supplement 1.

## 595    5 Conclusions

Our results confirm an isoprene-analogous behavior commonly attributed to MT emissions from QI and clarify and provide
more depth on several aspects regarding their $CO_2$-responsiveness: First, they corroborate earlier studies (Loreto et al., 1996b,
Staudt et al., 2001; Rapparini et al., 2004) showing that MT emissions become essentially inhibited under very high [$CO_2$] (Ci
>500 ppm), whereas smaller $CO_2$ variations (Ci: 200-500) affect little emissions. Second, contrary to the conclusions of Loreto
et al. (2001), $CO_2$-responsiveness is the same for all major MTs, regardless of tree chemotype. Third, the $CO_2$-responsiveness
is clearly temperature dependent. High leaf temperatures reduce the high-$CO_2$ inhibition of emissions, which can explain
seasonal differences in the $CO_2$-responsiveness of QI emission observed previously (Rapparini et al., 2004). Fourth, the leaf
emission factor measured at normal [$CO_2$] was positively related to plant growth and leaf photosynthetic activity. This initial

EGU Open Access

state affected the emission response to low $CO_2$, especially at high temperatures, but had no or little effect on the response to high $CO_2$, which was independently controlled by processes maintaining photosynthetic electron transport. Fifth, growth temperature rather than growth $CO_2$ influenced $CO_2$-responsiveness and this in the opposite way than did actual leaf temperature. Emissions of elevated temperature grown plants tended to be more inhibited by high $[CO_2]$ accompanied by a decline of photosynthetic electron transport and an increase of photoinhibition. Sixth, fitting the MEGAN algorithm to the whole data set of 30°-measurements resulted in a nonlinear response curve similar to that currently used for isoprene emission,

justifying its potential use in predicting the emission of de-novo-synthesized MTs under elevated $CO_2$. Finally, simulations estimating the annual BVOC release from QI leaves at double atmospheric $[CO_2]$ suggested that the observed $CO_2$-responsiveness is unlikely to offset the increase of leaf emissions from the expected warming. At higher $[CO_2]$ however, emission inhibition should become more relevant due to the non-linearity of the $CO_2$-response in interaction with increasing abiotic stress events.

More studies are warranted to establishing a sound knowledge on the variation of the high-$CO_2$ responsiveness of QI emissions, especially on adult trees under true field conditions (see e.g., Monson et al., 2016). In particular, it would be interesting to know whether its temperature sensitivity is stable during the course of the day and year or whether it changes due to recurrent changes in the leaf's source/sink balance and assimilate allocation pattern. $CO_2$-responsiveness of MT emissions might also be modulated by drought, the actual light conditions and photoperiod, as well as the plant's phenological status that all

influence the redox balance and partitioning of carbon precursors and energy inside the chloroplasts (Sun et al., 2012; Grote et al., 2014; Monson et al., 2016).

**Data availability.** The data used in this work are available from the corresponding author upon request (michael.staudt@cefe.cnrs.fr).


**Author contribution.** MS designed the experiment, analysed the data and wrote the manuscript; JD and JI performed the measurements; ND helped reviewing and editing the manuscript.

**Competing interests.** The authors declare that they have no conflict of interest.


**Acknowledgments.** We are grateful to all staff members of the CEFE institute platforms PACE and TE for their technical assistance, and in particular to Bruno Buatois and David Degueldre. We acknowledge the funding by the French National Research Agency.

**Financial support.** This study was funded by the project "ODORSCAPE" of the French National Research Agency (ANR15-CE02-010-01) with support of the LabEx CeMEB, an ANR "Investissements d'avenir" program (ANR-10-LABX-04-01).



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
