# Peer review of "Growth and actual leaf temperature modulate CO2-responsiveness of monoterpenes emissions from Holm oak in opposite ways"

_Biogeosciences, 2022_

## Author Response (AR1)

**Reply letter by the authors to the comments of the referees (bg-2022-142, 1st revision):**

We are very pleased that both referees find our work useful and encourage its publication in Biogeosciences. We thank referees and editor for the helpful suggestions to improve the manuscript and are pleased to propose a revised document. Revisions were made in all sections including Abstract, Figures, Conclusions, References and Supplementary information files. By far the largest revisions were made in the sections RESULTS and DISCUSSION, where major portions of text and even entire chapters were completely revised and moved as suggested referee 1. This sometimes makes the marked-up version of the revised manuscript difficult to read. In order to maintain readability, some revisions, such as replacing and moving figures, were not made in "track changes" mode.

In the following, we reply (in blue) point-by-point to the referee comments and explain our suggestions for improvements or rebuttals, some of which we have already posted in our previous comments on beginning of August. Please note that the line numbers mentioned in our answers refer to the "clean" revised manuscript (without annotated changes). Please also note that the images have been embedded in the text where they correspond to the actual content. They are not necessarily in the size they will appear in the final published article.

Referee 1
General Comment:
The authors did a very thorough investigation on a specific scientific question that certainly is of relevance for the evaluation of climate change impacts on biogenic emissions and feedbacks on air chemistry. In my opinion the experiment has been well set up and carried out. The interpretation is supported by a number of ancillary measurements so that some interesting ideas about the potential underlying mechanisms could be developed. Also, the authors revealed a well-founded knowledge about the topic and the relevant literature.
Answer: We are very pleased about these very positive and encouraging comments.

On the downside, I noticed that wording and style could be improved. Many sentences are inconveniently complicated or long and selected expression are often unfamiliar or imprecise. I would recommend to check, shorten, and involve an English native to improve the text.
Answer: We have revised the wording and style. However, the current revised version has not yet been reviewed by a native speaker or a professional language editor. If necessary, we will do so in the final accepted manuscript version.

Also some shifts between results and discussion sections and a better description of the equations used for sensitivity analysis should be considered at the appropriate places.
Answer: We agree and have made several text shifts within and between the RESULTS, DISCUSSION, and CONCLUSIONS sections, which we explain in more detail in our responses below. The MEGAN equation was placed in the main text (L 220).

Specific Comments:
Abstract
It seems unclear to me, what the cool and warm growth regimes look like. Indicating only the 5-degree difference is not sufficient. Compared with the quite extensively discussed results and conclusion, the description of the outcome is relatively meager.
Answer: We added the day/night temperatures of the temperature-growth regimes in the abstract (L 12). We have also revised and expanded the abstract text on the results and discussions about the $CO_2$ sensitivity of emissions, which should now be clearer and more in line with the wording in the other chapters (L17 ff).

Introduction

L65: I assume that MTs are not synthetized but only stored in resin ducts.

Answer: The terpenes in the resin ducts are synthesised in the glandular epithelium surrounding the cavity of the resin duct into which they are secreted (see e.g. https://nph.onlinelibrary.wiley.com/doi/full/10.1111/nph.15984 and references therein). Generally, the synthesis is particularly intense during resin duct development. Hence stored resin MTs are not synthesized in the photosynthetic parenchyma (photosynthetic source tissues) of leafs/needles and then transported into the resin ducts. However, MT synthesis in the resin duct epithelium may rely on photosynthates (essentially sucrose) provided by the photosynthetic parenchyma of source leaves (i.e. leaves that produce more photosynthates than they use for their own respiration and maintenance).

L76: superfluous 'very' (remove)
Answer: removed

L85: superfluous 'before' (remove)
Answer: removed

Description

There is a bit of a mix between description and discussion, check (e.g. L200-203)
Answer: We agree and removed the sentence "Previous studies showed that …".

Could you please indicate the equation used for emission factor reduction in MEGAN here (and not in the results as a caption text)?
Answer: Done (L220)

Define G400, A400
Answer: The definition of G and A and other variables of photosynthetic processes and leaf gas exchanges (and derivatives) are given in chapter 2.2 (L 133 ff). The different data normalization and transformations, the meanings of the corresponding abbreviations and suffixes and superscripts are explained in the section 2.4 (LL 205-218). We are aware that the use large number of abbreviations of ecophysiological variables and their derivatives makes our manuscript difficult to read. To provide more clarity, we have revised in 2.4 the explanations of the definitions, introduced equations, and improved some abbreviations. In addition, we have frequently recalled the meaning of abbreviations later in the text by using normal, unabbreviated wording and added the corresponding abbreviations in parentheses (see for example LL 316-318). The meanings of normalized data are also briefly re-explained in some figure captions (Figs 2, 3, 4, S1, S6 and S8).

Results

Figure 1: It is a bit irritating that the emission factor (per unit m2) should increase with the number of leaves. I see that the latter is meant as a growth indicator, which should, however, be better illustrated (e.g. final number of leaves? Number of leaves in the end of the growth period?)
Answer: Yes the total of leaves per plant was taken as a measure of the plant's growth performance. To better understand this, it should be noted that young potted QI plants in greenhouse culture do not have a fixed period of leaf growth as it occurs in the field. Under field conditions QI tress show typically only one leaf flush lasting from late spring to early summer (onset of drought), though under certain circumstances there can be a second one in the same year either from the buds formed in the spring and/or from dormant buds. In our

experiment, the QI saplings kept under none-stress conditions (well watered, no extreme temperatures) continued more or less growing in repeated cycles or even in indeterminate growth manner (central apex) until the end of the experiment. The acorns were potted at the same time and the number of leaves (plus few other morphological features) were determined at the end of the experiment in September. Also, there was no apparent difference in leaf size among the four growth treatments with only moderate differences in specific leaf weight (LMA). These facts allowed us to consider the number of leaves as a proxy for foliage growth. The plants were not harvested after the experiment, because we wanted keeping them alive for eventual additional measurements. Finally these were not made, due to the lack of time, manpower and because the plants had to leave rapidly the greenhouse compartments.

Figure 2: Better use the same design for Ci in each of the graphs (i.e. that which shows relative NPQ)
*Answer: Scaling corrected*
Figure 3: You probably mean key relations instead of key correlations. Actually, I have difficulties to see understand both, the explanations of how this is calculated and the reason why it has been done.
Answer: We replaced "key correlations" by "key relations" in the captions.
The scope and principle of Pearson correlation analysis is briefly explained on L231. It simply analysis the linear relationships among quantitative variables. The result of the correlation is given as Pearson correlation coefficient R, which ranges between -1 and +1 and provides information about the direction and strength of the linear relationship (see Table S3 in Supplement 2), or as determination coefficient $R^2$ (0-1) (see Figures 1 and 3), which is a measure of goodness of fit explaining the proportion of variance explained by the model (linear relationship in case of Pearson). The totality of the results are typically shown in matrices with numerical values (ex. R), colors (heat maps) or scatter plots. Scatter plots are particularly informative, because they show the input data in its original form and its distribution (possible presence of outliers, clusters, tendencies for non-linear relationships). However, when the Pearson analyses involve a large number of variables, as in the present study, matrices showing scatter plots with all variable combinations are too messy. Therefore we show in Figures 1 and 3 only the most interesting scatter plots with the key variables of interest of our study, which are the emission factor EF, the mean relative emissions at low $CO_2$ ($\mu$ E<400 E400-1) and at high $CO_2$ (($\mu$ E>400 E400-1). The results of other correlations are mentioned in the text with their corresponding R and P values and the totality is summarized in the R matrices of Table S3. The latter were revised and improved by adding an coloration distinguishing the direction and the significance level of the correlations. In addition, we presented a subset of the results in extra matrices (Table S3a), in which only variables are shown that are relevant for understanding EF variability.

L364-366: The difference between the explanations for the two different responses to temperature are unclear. Rephrase and consider to elaborate the arguments.
Answer: These explanations were removed from the section RESULTS along with the former Fig. 5. The possible mechanisms underlying the emission responses to $CO_2$ and the influences of test and growth temperature are now exclusively discussed in the section DISCUSSION (and summarized in the CONCLUSIONS).

L371ff: Should this really be one figure caption? Generally, I expect a short, clear and consistent description of what I see. This is violated at least since line 376. Instead, take care that the abbreviations are all clear (e.g. chloro, growth?). It could also be considered to use

this figure as a basis for discussion and put into chapter 4, possibly in several stages in order to better support the reasoning in the different chapter.

Answer: This figure was thought to provide readers an overview and summary of the outcome of the Pearson correlation analyses (see our answer ebove). However because it is hard to read and difficult to place in the main text we removed it as suggested by Referee 2. Nevertheless we revised this figure and suggest to keep it as supplementary figure S7 in supplement 2 as a support of the results shown in the correlation matrices (Table S3 in Supplement 2) and scatter plots of Figures 1 and 3.

Discussion

What I am missing is a discussion in how far the results can be assumed general findings or are specific for Quercus ilex? Is it likely that conifers, evergreens, broadleaves or Mediterranean plants react similar? Do you think the BVOC emission groups should then be differentiated by their degree of genetic relatedness or to site conditions typical for the species?

Answer: This is indeed an important point with respect to emission modeling and inventories. As mentioned in the manuscript, our results show a strong similarity with isoprene emissions indicating that the MEGAN algorithms with current coefficients used for isoprene emissions is also valid for monoterpene emissions from Holm oak and perhaps to other monoterpene emitters, where emissions are directly linked to their de-novo synthesis in photosynthetic tissues. However, our results also shows that Holm oak has specific characteristic that together with unknown factors and assumptions made in our simulations limit the transferability of the results ("extrapolation power"). In the revised manuscript, many of these points are now addressed in the chapter 4.3 of the section Discussion (Relevance for predicting MT emissions), which was completely changed (please see our answer below).

L513ff: With the summary here, the paragraph tends to be lengthy and repetitive. I would suggest to take the essence from this paragraph to the conclusions (and delete it here).

Answer: We agree and integrated a part of this paragraph in the paragraph before (LL 560-567).

L550ff: Here, for the first time if I am not mistaken, the authors declare that they also run some simulations to test the sensitivity of the found mechanisms. While I am not against such exercises, this comes as a surprise and should have been mentioned and described before (and shorten it here). Also Fig. 6 is a result and only part of its description belongs into discussions.

Answer: We fully understand that this part of the DISCUSSION was surprising after the previous part dealing essentially with ecophysiological aspects. The idea of this chapter was to run some simulations to assess the relative importance of the observed $CO_2$-inhibition of emission at an annual scale in a kind of sensitivity analysis. These simulations are based on the results obtained in the present study that were combined with other data and results and modeling approaches from previous published studies. Hence this chapter comprised a mix of results and discussion and therefore we had placed in at the end of the section DISCUSSION. In the revised version we follow the suggestions of referee 1 and moved large parts of it to a extra chapter in the section RESULTS entitled "Implications for predicting future MT emissions from Holm oak" (chapter 3.4, L359ff), in which we also integrated the results of the fitting of the data to the MEGAN algorithm (Figure 4). In this new chapter, we focus mainly on the results of the simulations shown in Figure 5 (former Figure 6), which has been extended showing now the results of all 96 simulations. The limitations of the results shown

in 3.4 (uncertainties, unaccounted factors, special characteristics of Holm oak…) are now discussed in the chapter 4.3 "Relevance for predicting MT emissions" (L 569ff).
To keep the new RESULTS chapter 3.4 in length, we have kept the description of how the simulations were performed brief. For readers interested to know more about we generated a new supplementary information file (supplement 3), in which we describe our simulation approach in detail. In this Supplement 3, we also placed all supplementary tables and figures related to chapter 3.4.

Conclusion
L599: concentrations instead of variations; "hardly effect emissions" or "affect emissions only marginally" or similar instead of "affect little emissions". (good example for wrong wording)
Answer: Thank you very much for the concrete examples, which helped us to improve the wording.

L615ff: Missing knowledge as well as stating additional references is not something, that should be put into a conclusion. Please consider to shift it towards the discussion.
Answer: We revised the Conclusions accordingly. The new version concentrates on the main findings and conclusions in a "non bullet point" style without references. Some of the missing knowledge (with references) stated in the old version was shifted to the end of the discussion chapter 4.3.

Referee 2
**Growth and actual leaf temperature modulate CO2 -responsiveness of monoterpene emissions from Holm oak in opposite ways**
The manuscript describes a greenhouse experiment where the effects of elevated CO2 and growth temperature on holm oak leaf scale monoterpene emission rates are assessed. This is very relevant research topic already for decades, and the authors manage to scrutinize the experiment in a way that they can eventually conclude novel and interesting results.
The monoterpene emission responses to elevated CO2 and temperature were decoupled. Clear differences between cool- and warm-grown plants could be seen, the latter being more sensitive to CO2 inhibition. Contrasting this, a lower actual measurement temperature seemed to lead to larger CO2 inhibition compared to measurements at higher (35C) temperatures.  This is rather surprising when the temperature difference is only 5C. The authors explain this with the leaf energy balance, similarly as has been shown for isoprene. Still, some explanations of the seemingly rather small temperature difference should be interesting for readers. In contrast, growth CO2 had no significant effect on emission CO2 sensitivity, although it promoted plant growth and the leaf's emission factor.
The methods are well designed and elegantly used. Several different normalisation methods are used for assessing the uncertainties related to plant chemotype, growth conditions and measurement conditions. Finally, the obtained non-linear responses are used to upscale the short term impacts to annual emission dynamics using the MEGAN algorithm.
Overall, the ms represents an elegant experiment and is well compiled. It could be revised by removing some of the speculations and using the figures more directly to show the reader the main results, this would lead to significant shortening and clarifications of the main messages.
Answer: Thank you very much for these very positive words. We have considered all advices during our revision. Parts of the sections RESULTS and DISCUSSION have been completely revised and reorganized. Some lengthy and repetitive information have condensed or removed (please see our answers to the comments of referee 1).

Some linguistic errors and typos should be corrected, and a few other aspects could be clarified in the manuscript:
how old were the measured leaves, were they of same age? what part of the canopy?
Answer: All leaves were mature leaves of the current year. The age of the leaves in months is not known. For practical reasons, leaves were selected that were not too small and were at the end of the shoots so that they could be accommodated in the LiCOR leaf chamber and covered the entire chamber surface.

how tall were the saplings?
Answer: the height of the saplings ranged between 15 and 70 cm with variable branching.

what was their rooting size?
Answer: The size of roots or any measure requiring the harvest of the plants were (unfortunately) not made at the end of the experiment.

emission measurements: how many adsorbent tubes per CO2 level and leaf?
Answer: Only one VOC sample per CO2 level so that the VOC sampling phase and consequently the duration of the entire CO2 ramping was not too long. Overall, we had very few sample losses due to errors in the GC-MS analysis. In contrast, we lost entire CO2 ramping series because the leaf was injured or even detached (the petiole of Q. ilex leaves is very short).

was the humidity of incoming air controlled?
Answer: The humidity of the incoming air was not controlled (H2O-scrubber was not used).

Supplementary table 1 has remnants of non-english origin (mars)
Answer: corrected.

Figure 5 is an overview of the correlation network, but is does not really clarify the results and is almost impossible to read. I recommend removing it. However, I was missing a multivariate analysis where the combined effects of temperature and CO2 levels could have been assessed.
Answer: We agree and shifted the figure to the supplementary material (now Figure S7 in Supplement 2). With regard to multivariate analyses, we considered the possible use of principal component and discriminant analyses for our data. However, we felt that these tools would add little or no value in terms of clarity, but would rather make the manuscript longer. We believe that the scatter plots shown in Figures 1 and 3, together with the improved correlation matrices (Table S3 in Appendix 2) and the detailed description in the text, provide a good overview of the main relationships between variables.

**Final remark to the referees:**
We are grateful to the two reviewers for their careful reviewing of our manuscript and their stimulating comments. Although some additional corrections may still be necessary for final acceptance, we believe that we have addressed all of the reviewers' comments and suggestions and that the "minor" corrections we have made have significantly improved the structure and readability of our manuscript. We hope that the reviewers and the editor share our opinion.

With kind regards,
Michael Staudt (on behalf of the co-authors)